# Menstrual Blood Stem Cells-Derived Exosomes as Promising Therapeutic Tools in Premature Ovarian Insufficiency Induced by Gonadotoxic Systemic Anticancer Treatment

**DOI:** 10.3390/ijms25158468

**Published:** 2024-08-02

**Authors:** Mariana Robalo Cordeiro, Ricardo Roque, Bárbara Laranjeiro, Carlota Carvalhos, Margarida Figueiredo-Dias

**Affiliations:** 1Faculty of Medicine, Gynecology University Clinic, University of Coimbra, 3000-548 Coimbra, Portugal; barbaralaranjeiro@gmail.com (B.L.); carlota.carvalhos@hotmail.com (C.C.); marg.fig.dias@gmail.com (M.F.-D.); 2Portuguese Institute of Oncology of Coimbra, Medical Oncology Department, 3000-075 Coimbra, Portugal; roque.jricardo@gmail.com

**Keywords:** extracellular vesicles, menstrual blood-derived stem cells, regenerative medicine, premature ovarian insufficiency, cancer

## Abstract

Gonadotoxicity resulting from systemic and locoregional cancer treatments significantly threatens women’s reproductive health, often culminating in premature ovarian insufficiency. These therapies, particularly alkylating agents and ionizing radiation, induce DNA damage and apoptosis in ovarian follicles, leading to infertility, amenorrhea, and estrogen deficiency, which exacerbate risks of osteoporosis and cardiovascular diseases. Existing fertility preservation methods do not prevent immediate ovarian damage, underscoring the need for innovative protective strategies. Menstrual blood-derived stem cells (MenSC) and their extracellular vesicles (EV) present promising regenerative potential due to their therapeutic cargo delivery and pathway modulation capabilities. Preclinical studies demonstrate that MenSC-derived EV ameliorate premature ovarian insufficiency by inhibiting granulosa cell apoptosis, promoting angiogenesis, and activating pivotal pathways such as SMAD3/AKT/MDM2/P53. However, comprehensive research is imperative to ensure the safety, efficacy, and long-term effects of MenSC-derived EV in clinical practice. In this review, we update the current knowledge and research regarding the use of MenSC-derived EV as a novel therapeutic weapon for ovarian regeneration in the context of gonadotoxicity induced by systemic anticancer treatment.

## 1. Introduction

Gonadotoxicity from systemic and locoregional cancer treatments significantly jeopardizes ovarian function and reproductive health in women. These therapies, while efficacious in targeting tumor cells, cause DNA damage and apoptosis in ovarian follicles, often leading to premature ovarian insufficiency (POI) [1,2]. Alkylating agents and ionizing radiation are particularly harmful, resulting in follicular depletion that manifests as estrogen deficiency, amenorrhea and infertility [2,3]. Moreover, these conditions further increase the likelihood of osteoporosis and cardiovascular diseases, severely impacting a patient’s quality of life. 

The extent of gonadotoxicity is shaped by variables like patient age, chemotherapy type and dosage, and treatment duration [3]. Current fertility preservation methods, like oocyte and embryo cryopreservation, offer future fertility options but do not prevent immediate ovarian damage. Hence, the development of innovative strategies aimed at protecting ovarian integrity in the context of cancer treatment is imperative.

Recent advances in regenerative medicine have identified stem cell-based therapies as potential interventions for ovarian regeneration. Among the myriad of stem cell sources, menstrual blood-derived stem cells (MenSC) have gathered considerable interest due to their non-invasive obtaining ways, robust proliferative capacity, and extensive differentiation potential [4]. MenSC, first characterized in 2007, represent a unique population of endometrial stem cells harvested from menstrual blood [5]. The exclusive marker expression profile confers MenSC with a remarkable ability to differentiate into diverse cell lineages, such as adipocytes, osteocytes, cardiomyocytes, neurons, respiratory epithelial cells, endothelial cells, myocytes, hepatic cells, germ-like cells, pancreatic cells, and ovarian tissue-like cells [5].

Extracellular vesicles (EV), encompassing exosomes and microvesicles, are membrane-bound particles secreted by cells that facilitate intercellular communication. These vesicles encapsulate a variety of bioactive molecules, including proteins, lipids, and nucleic acids, which can modulate the behavior and function of recipient cells [6]. MenSC-derived EV have been noted for their regenerative properties, largely attributed to their ability to deliver therapeutic cargo to target cells and influence critical signaling pathways involved in cellular survival, proliferation, and differentiation. In the domain of systemic anticancer treatment-induced gonadotoxicity, MenSC-derived EV offer a novel and promising therapeutic avenue [7]. Emerging evidence suggests that these EV can significantly ameliorate POI and enhance ovarian function by inhibiting granulosa cell apoptosis, promoting angiogenesis, and modulating key molecular pathways [8]. 

This article aims to provide a comprehensive review of the therapeutic potential of MenSC-derived EV in the context of systemic anticancer treatment-induced gonadotoxicity, exploring their molecular mechanisms of action and discussing their prospective applications in clinical practice after proven safety and efficacy in multicentric clinical trials.

## 2. Gonadotoxicity Induced by Systemic Anticancer Treatment in Women

Systemic and locoregional cancer treatments may affect women’s fertility in various ways: reduction in the primordial follicle pool (i.e., ovarian function reduction) [9]; hormonal imbalance; or through anatomical or functional changes in the reproductive system. Reduced ovarian function can lead to premature ovarian insufficiency, which occurs before the age of 40 and is characterized by oligo/amenorrhea lasting 4 months or more and follicle-stimulating hormone (FSH) levels of >25 IU/L on two occasions, 4 weeks apart [2,9]. Retrospective cohorts show that around 30% of pre-menopausal women will suffer from POI after chemotherapy [10,11] and up to 90% of post-pubertal patients will experience a reduction in ovarian reserve at 50 months, represented by anti-Müllerian hormone (AMH) levels below 0.5 ng/mL [10].

Many women will experience amenorrhea in a transient way during chemotherapy (ChT); however, resumption of menses months after ChT does not mean that infertility or POI will not occur [9]. Various factors influence the outcomes of a patient’s ovarian function after systemic anticancer treatment, such as age, pretreatment ovarian function, type of malignancy and type of treatment received [9,11]. Also, cumulating evidence suggests that the existence of cancer per se negatively impacts ovarian function before any treatment [1,12]. However, irrespective of any of those variants, fertility counselling should be offered before treatment, as well as fertility preservation for those interested, including pediatric and adult patients with cancer [2,9,13]. 

Regarding chemotherapy, multiple mechanisms contributing to ovarian damage have been reported. Apart from the direct apoptotic death of primordial follicles, a “burn-out” mechanism consisting of an induction in follicle activation followed by follicle loss, which is associated with an indirect effect on follicle pools by damaging the surrounding stroma and disrupting blood supply, is the most reported ovarian damage mechanism in animal models and in humans [3,14,15]. Within ChT agents, alkylating agents like cyclophosphamide seem to have a greater effect on ovarian function because of their cell cycle-independent mechanism that affects both growing follicles and oocytes [9,15]. The amenorrhea risk can be higher than 80%, particularly when polychemotherapy is used in women aged 40 or over. Such combinations are particularly common in the treatment of breast and hematological malignancies [9]. Likewise, commonly used across different types of solid cancers, platinum compounds like cisplatin or carboplatin are cell cycle-independent and have an intermediate risk of causing amenorrhea, as well as doxorubicin [3,16]. Antimetabolites like 5-FU or methotrexate, vinca alkaloids and other anthracyclines like bleomycin are of lower risk (beneath 20%) [3]. Taxanes seem to have a mild gonadotoxic effect; however, the true clinical effect is still unknown [17].

Hormone therapy, namely tamoxifen, used in premenopausal women with hormone receptor-positive breast cancer, seems to contribute to chemotherapy-induced amenorrhea and has shown to negatively impact the probability of pregnancy in a population-based cohort [18,19,20]. On the contrary, some authors hypothesize a potential protective role of tamoxifen since it does not impact the values of AMH. Therefore, the true clinical impact of tamoxifen on ovarian reserve remains unknown [20]. 

In the era of targeted treatments, the impact of immunotherapy and other cancer-targeting antibodies on fertility is still unclear [9,21]. Theoretically, commonly used anti-cancer antibodies disrupting the epidermal growth factor receptor 2, anti-HER2, like trastuzumab and pertuzumab could disrupt oocyte maturation; immunotherapy-induced endocrinopathy like hypothyroidism and hypophysitis can lead to reduced gonadotropin secretion and ovarian insufficiency; and poly (ADP-ribose) polymerase inhibitors like olaparib and niraparib can induce follicle loss through lack of DNA repair and subsequent genomic instability [21]. Sub-analysis of phase III trials, however, showed that anti-HER2 antibodies and even anti-HER2 tyrosine kinase inhibitors (TKIs), like lapatinib, do not seem to impact the rate of amenorrhea, even when linked with a chemotherapy molecule, like ado-trastuzumab emtansine (TDM-1), an antibody-drug conjugate [22,23,24,25]. 

TKIs have a wide variety of effects depending on the kinases they inhibit, and their effect on ovarian function is unknown but apparently reversible. Angiogenic disruption and potential ovarian dysfunction caused by antiangiogenic-targeted treatments, like bevacizumab, and anti-VEGF TKIs cannot be excluded [9,21,25]. 

## 3. Challenges in POI Induced by Systemic Anticancer Treatment 

Premature ovarian insufficiency is a serious and impactful diagnosis for a woman. It is a condition with severe medical, psychological, and reproductive consequences. Besides menstrual disturbances and infertility, it may cause numerous health problems, mainly due to estrogen deficiency, throughout women’s lives [26].

Menopausal symptoms extend beyond vasomotor symptoms like hot flushes and night sweats, including, in the short-term, mood swings, sexual dysfunction, sleep impairment, genitourinary symptoms, and an overall decrease in quality of life [26,27]. In the long term, premature ovarian insufficiency might increase the risk of cardiovascular disorders, osteoporosis, some degree of cognitive deterioration, and lower life expectancy. In cancer survivors, POI-related symptoms are frequently more severe and their consequences often receive little attention, as follow-up care is focused predominantly on surveillance for cancer recurrence. While the standard of care for POI is menopause hormone therapy (MHT), there are few cases where it is contraindicated [28]. Hence, MHT management should be tailored to the individual, considering the patient’s age, cancer type, time since diagnosis, severity of symptoms, comorbidities, risk factors for chronic disease, and patient preferences.

Until recent years, ovarian senescence in POI patients was considered an irreversible natural aging process [27]. Currently, there has been a switch in this paradigm and novel therapies trying to restore ovarian function after POI diagnosis are under promising investigation. Furthermore, there are several studies, mainly in animal models, searching for molecules able to preserve ovarian function after systemic anticancer treatment; however, to date, none are available for clinical use (Table 1) [9,15]. Sphingosine-1-phosphate and tamoxifen have also been studied in human xenograft models and shown to reduce follicular loss and apoptosis in mice treated with cyclophosphamide and doxorubicin [29,30]. However, none of these molecules, apart from gonadotropin-releasing hormone agonists (GnRHa), have reached human studies [31]. 

i.Gonadotropin-releasing hormone agonists

Gonadotropin-releasing hormone (GnRH) is a decapeptide synthesized in the hypothalamus that targets GnRH receptors located in the anterior pituitary [31,32]. This interaction promotes the secretion of luteinizing hormone (LH) and follicle-stimulating hormone (FSH), which subsequently induce the synthesis and release of testosterone in men and estrogen in women, particularly from the ovaries in the latter ones. GnRH is typically secreted in a pulsatile manner, which is modulated by the circulating concentrations of estrogen in women. Administering GnRH agonists results in an initial rise in sex hormones; however, sustained non-pulsatile exposure suppresses hypothalamic LH and FSH production, leading to decreased levels of estrogen [32]. 

After the encouraging foundational experimental studies by Ataya et al. in 1980, GnRHa, like goserelin, have been largely studied as an approach to ovarian protection during chemotherapy, resulting in several randomized trials and meta-analyses, mainly conducted in breast cancer patients [31,33]. The POEMS trial found a higher rate of pregnancy in women treated with goserelin during breast cancer adjuvant or neoadjuvant chemotherapy, further supported by a lower rate of POI incidence and higher resumption of regular menses found in further meta-analyses [33]. Despite its wide availability for clinical use in these settings, GnRHa should not substitute traditional fertility preservation methods whenever desired by patients and if available.

ii.Platelet-rich plasma therapy

Platelet-rich plasma (PRP) is an autologous plasma with a higher-than-baseline concentration of platelets, obtained through the centrifugation of patient’s own blood [34]. PRP is abundant in growth factors and cytokines that play critical roles in cell differentiation and proliferation, tissue regeneration, and angiogenesis. 

Growth factors in PRP include platelet-derived growth factor (PDGF), epidermal growth factor (EGF), basic fibroblast growth factor (bFGF), vascular endothelial growth factor (VEGF), transforming growth factor-beta 1 (TGF-β1), insulin-like growth factor 1 (IGF-I), and hepatocyte growth factor (HGF) [34]. These growth factors are able to activate oocytes by producing a substantial amount of cellular growth factors through the combined action of autologous platelets and gonadotropins, thereby creating favorable conditions for the differentiation and development of primordial follicles [35]. Recent studies have indicated that EGF can activate the small GTP-binding protein CDC42 and its phosphorylation. Yan et al. observed significantly elevated levels of FOXO1a and Akt expression in CDC42-treated follicular fluid, suggesting that the PI3K/AKT signaling pathway in ovarian follicular fluid is activated due to CDC42, which promotes follicular differentiation and development [36]. HGF can also significantly influence ovarian development. Mi et al. treated murine ovaries with HGF and found a notable increase in KITL expression in HGF-treated ovaries, with a corresponding decrease in phosphorylated AKT and FOXO3a expression levels upon the addition of KITL inhibitors [36]. This indicates that HGF activates the PI3K/AKT pathway by upregulating KITL expression, thereby activating primordial follicles. Similarly, PDGF activates the PI3K/AKT pathway, enhancing ovarian function in POI patients. El Bakly et al. demonstrated that PRP can stimulate follicular growth by activating the mTOR signaling pathway in a POI mouse model via intraovarian PRP injection [36]. 

Since 2018, human studies have been performed using intraovarian PRP injection to evaluate its ability to restore ovarian function. Pantos et al.’s results seem very promising, including a reduction in FSH serum levels and spontaneous pregnancies within 2 to 6 months in a few cases after PRP intraovarian injection [36]. Further research by Sfakianoudis et al. also observed a 60% restoration in menstruation, improved hormone serum levels, and three spontaneous pregnancies all ending in live births [36]. In 2020, Cakiroglu et al. also showed a significant increase in the number of ovarian follicles and AMH serum levels in most POI patients submitted to intraovarian PRP injection [36]. 

Encouraged by the aforementioned results, many researchers have studied the potential role of PRP in the restoration of ovarian function in cyclophosphamide(Cy)-induced POI rats. Ozcan et al. observed an increased number of primordial, primary, and antral follicles, and histologically preserved ovarian architecture, and reduced follicular degeneration in PRP-treated Cy-exposed rats compared to the untreated group [37]. Similar results were found by Dehghani et al., including a decreased expression of apoptotic markers, such as caspase-3, and higher expression of proliferating cell nuclear antigen (PCNA) in PRP-treated rat ovaries [35]. In 2019, Huang et al. comprehensively evaluated the combined effect of granulocyte colony-stimulating factor (G-CSF)-mobilized peripheral blood mononuclear cells (PBMCs) and PRP on ovarian function in a Cy-induced POI rat model [38]. Apart from the improvement in follicular development, reduction in apoptosis and oxidative stress, and improved ovarian histology in the PRP-treated group, their results showed a promising restoration of ovarian endocrine function, with increased serum estrogen (estradiol, E2) and AMH levels [38].

It is important to note that the majority of these studies involved, at most, small case series; these were mainly performed in animal models and lack follow-up data. Thus, whether PRP actually improves ovarian function is still unknown. Furthermore, extreme caution must be taken due to the putative but not confirmed risk of PRP growth factors as mutagenic vehicles and capable of inducing tumor cell proliferation.

iii.Mesenchymal stem cell therapy

Stem cells have become a vital component of regenerative medicine. Mesenchymal stem cells (MSC) are pluripotent stem cells originating from the mesoderm layer, primarily sourced from adult tissues such as bone marrow and fat, as well as from fetal tissues like the umbilical cord and placenta [4]. MSC can be induced to differentiate into various types of mesenchymal tissues, including bone, cartilage, fat, and heart muscle, under specific physiological conditions throughout the organism [4]. The unique properties of MSC make them ideal for cell-based therapy [39].

In 2019, Liu et al. published a study in which the bilateral ovaries of female mice were burned with 10% hydrogen peroxide to establish a POI model and subsequently transplanted with human amniotic mesenchymal stem cells (hAMSC). They observed that estrogen levels, ovarian index, fertility rate, and population of follicles at different stages were significantly increased after the treatment [40]. These results suggested that hAMSC transplantation can improve injured ovarian tissue structure and function in oxidatively damaged POI mice.

Similarly, Mohamed et al. successfully restored endocrine and exocrine ovarian functions after administering umbilical cord blood mesenchymal stem cells (UCMSC) to the chemotherapy-damaged ovaries in mice, further demonstrating the ability of MSC to restore fertility in female cancer survivors with POI [41].

Previous studies indicate that adipose-derived stem cell (ADSC) therapy is restricted by insufficient settlement of transplanted cells in the target tissue. In efforts to try to overcome this problem, Su et al. demonstrated that collagen scaffolds increased the retention of ADSC in ovaries and contributed to long-term restoration of ovarian function [42]. ADSC transplantation on collagen scaffolds improved the fertility of rats after tripterygium glucoside-induced ovarian damage [42]. Thus, collagen/ADSC transplantation may represent another promising candidate for POI treatment.

In 2020, Yang et al. studied the effect of bone marrow mesenchymal stem cells (BMSC) and BMSC-derived exosome transplantation in cyclophosphamide (CTX)-exposed rats. BMSC-derived exosomes delivered miR-144-5p, which were then transferred to cocultured CTX-damaged granulosa cells (GC) [43]. The conclusions revealed that overexpression of miR-144-5p in BMSC showed efficacy against CTX-induced POI and showed a decrease in GC apoptosis and an improvement in cell repair [43]. Globally, the BMSC and BMSC-derived exosome transplantation showed potential in significantly recovering the estrus cycle, increasing the number of basal and sinus follicles in POI rats, increasing estradiol and AMH levels, and reducing FSH and LH serum levels [43].

Menstrual blood-derived stem cells (MenSC) were initially identified in 2007 and are recognized as a valuable source of endometrial stem cells extracted from menstrual blood. MenSC predominantly exhibit the expression of certain cell surface markers, including CD9, CD13, CD29, CD41a, CD44, CD59, CD73, CD90, and CD105, while lacking the expression of markers such as CD19, CD34, CD45, CD117, CD130, or human leukocyte antigen-DR isotype (HLA-DR) [5]. Additionally, some studies have indicated the presence of embryonic and intracellular multipotent markers, such as OCT-4, c-kit proto-oncogene/CD117, and stage-specific embryonic antigen-4 (SSEA-4), which are typically absent in MSC derived from other tissue sources. MenSC can differentiate into adipocytic, osteogenic, cardiomyocytic and neuronal lineages, as well as respiratory epithelial, endothelial, myocytic, hepatic, germ-like, pancreatic cells and ovarian tissue-like cells [5]. These stem cells have remarkable and unique properties, namely a very high proliferative rate with genomic stability, immunomodulatory and anti-inflammatory capabilities, and paracrine effects involved in tissue and organ regeneration, alongside their important role in angiogenesis [5,39]. Nevertheless, a large number of experimental and clinical studies revealed that most of MSC-based immunomodulatory effects, including MenSC, were attributed to the immunoregulatory properties of MSC-sourced secretomes [44].

The pivotal concern in applying stem cells in practice seems to be adjusting the dose and method of administration to optimize outcomes. Overall, MSC, namely MenSC, represent an alternative and extraordinarily promising strategy for POI therapy, particularly in systemic anticancer therapy-based gonadotoxicity, and all the foreseen results are very encouraging.

## 4. Clinical Potential of Extracellular Vesicles Derived from MenSC on Premature Ovarian Insufficiency

First discovered in the early 1980s, extracellular vesicles (EV) are nanoscale, lipid bilayer-enclosed particles secreted by cells into the extracellular milieu. These vesicles are pivotal mediators of intercellular communication, encapsulating and transferring a diverse repertoire of bioactive molecules, including proteins, lipids, and various nucleic acids such as mRNA, microRNA, and DNA [6]. EV are classified primarily based on their biogenesis and size into three main classes:i.Exosomes: These are small vesicles ranging from 30 to 150 nanometers in diameter. They originate from the endosomal system, specifically from the intraluminal vesicles formed within multivesicular bodies (MVB). Exosomes are released into the extracellular environment when MVB fuse with the plasma membrane [6].ii.Microvesicles: Also referred to as ectosomes, these vesicles vary from 100 to 1000 nanometers in diameter. They are generated through the outward budding and fission of the plasma membrane [6].iii.Apoptotic bodies: These are relatively larger vesicles, ranging from 500 to 2000 nanometers in diameter, formed during apoptosis. They encapsulate cellular debris and apoptotic fragments [6].

EV can be released from various cell types from prokaryotic to mammalian cells and stem cells. Hence, they are ubiquitous in different body fluids, including blood, urine, saliva, and cerebrospinal fluid [6]. These vesicles are deemed optimal therapeutic agents due to their non-viable and non-replicative nature, their ability to shield their bioactive cargo from degradation, and their capacity to maintain potency and functional integrity throughout processes of handling, storage, and administration [6]. Therefore, they exhibit a plethora of favorable therapeutic properties mediated by their diverse cargo of biomolecules (Figure 1) [45].

MenSC-derived EV are typically classified as exosomes or small EV (sEV). Under transmission electron microscopy (TEM), MenSC-derived EV exhibit a characteristic round or cup-shaped morphology, indicative of well-preserved vesicular structures. These EV express specific tetraspanins such as CD9, CD63, and CD81, which are commonly used markers for exosomes, and other distinctive markers including heat shock proteins (HSP70, HSP90) and the endosomal sorting complex required for transport (ESCRT) protein TSG101 [7,45]. They do not express markers such as calnexin and Rab5, which are associated with intracellular compartments, confirming their extracellular origin and purity [7,44,46]. MenSC-derived exosomes were found to also contain bioactive molecules, including microRNAs (miRNAs), proteins, and lipids. Key miRNAs identified in MenSC exosomes included miR-21, miR-126, and miR-146a, which are known to regulate cell survival, angiogenesis, and immune responses [7,44]. These exosomes also modulated critical signaling pathways in recipient granulosa cells. For instance, miR-21 is known to target PTEN and activate the PI3K/AKT pathway, promoting cell survival and proliferation. Moreover, miR-126 enhances angiogenesis by targeting SPRED1, thereby activating the VEGF pathway [7,44].

Research on exosomes derived from MenSC has elucidated their significant potential in modulating cytokine levels, which are crucial mediators of immune responses. Chen et al. observed that MenSC-derived exosomes significantly decreased levels of TNF-α, IL-1β, and IL-6 in an in vitro mouse AML12 hepatocyte cell line model [47]. These cytokines are key players in the NF-κB signaling pathway, which is central to the inflammatory response. As mentioned, by delivering miRNAs such as miR-21 and miR-146a, these exosomes inhibit the translation of mRNA encoding these cytokines, thereby dampening the inflammatory response. Furthermore, these cytokines activate the SMAD2/3 signaling pathway, promoting immune tolerance and resolving inflammation [46,47]. The immunomodulatory capacity of MSC-derived, including MenSC-derived, exosomes is highlighted by their interactions with various immune cells, significantly influencing their function and activity. Dai et al. demonstrated that these exosomes inhibit T-cell activation by transferring miRNAs such as miR-155 and let-7b, targeting transcripts of key activators like STAT3 and NFAT5, resulting in reduced T-cell proliferation and cytokine production [48,49]. Furthermore, these exosomes increased the population of regulatory T-cells (Tregs) in a mouse model of autoimmune encephalomyelitis, enhancing immune tolerance. He et al. showed that MSC-derived exosomes promote macrophage polarization from the pro-inflammatory M1 phenotype to the anti-inflammatory M2 phenotype by delivering miRNAs like miR-223 and miR-146b, which target NF-κB and STAT1 [50]. Housseini et al. reported that these exosomes prevent dendritic cell maturation by transferring miRNAs and proteins that downregulate co-stimulatory molecules and cytokines required for T-cell activation, leading to reduced activation of naïve T-cells. [51] Additionally, MenSC-derived exosomes have been shown to affect apoptosis-regulating proteins such as Bax, Bcl-2, and caspase-3 [52]. These exosomes can increase the expression of the anti-apoptotic protein Bcl-2 while decreasing the expression of the pro-apoptotic protein Bax and the activity of caspase-3, leading to enhanced cell survival and reduced apoptosis [52]. MenSC-derived EV offer a promising multifaceted therapeutic approach for the treatment of POI, leveraging their diverse molecular cargo and intricate signaling mechanisms to promote ovarian regeneration, reduce inflammation, enhance angiogenesis, mitigate oxidative stress, and restore hormonal balance (Figure 2) [8,53,54,55,56]. However, the paucity of research concerning this domain underscores the need for careful interpretation of the limited data available, and any clinical correlations drawn must be approached with a heightened level of scrutiny to ensure their validity and reliability.

Recent in vitro studies have elucidated both the direct and indirect molecular mechanisms by which EV derived from MSC affect estrogen secretion. Directly, MSC-derived EV enhance estradiol production in granulosa cells by upregulating the expression of aromatase (CYP19A1), the key enzyme responsible for the biosynthesis of estrogens from androgens. Proteomic and transcriptomic analyses have revealed that these EV contain specific microRNA, such as miR-223 and miR-125b, and proteins that target and modulate signaling pathways involved in steroidogenesis, including the cAMP/PKA and MAPK pathways. For instance, miR-223 is known to downregulate the expression of nuclear receptor co-repressor 1 (NCoR1), thereby relieving its inhibitory effect on aromatase expression and promoting estrogen synthesis [49,57]. Additionally, miR-125b targets and suppresses the expression of pro-apoptotic gene BAK1, enhancing granulosa cell survival and thus indirectly supporting estrogen production [58]. Indirectly, MSC-derived EV contribute to the enhancement of estrogen secretion by improving the proliferation and reducing the apoptosis of granulosa cells, which are essential for sustained estrogen production. EV carry bioactive molecules such as growth factors (e.g., IGF-1, TGF-β) and anti-apoptotic miRNA (e.g., miR-21, miR-214) that activate pro-survival signaling pathways, including the PI3K/AKT and MAPK/ERK pathways [48,57,58,59]. This leads to increased granulosa cell viability and function under stress conditions, such as oxidative stress or aging, which otherwise impair cellular performance [57,59]. Additionally, studies have shown that EV can modulate the expression of anti-apoptotic proteins such as Bcl-2 and Bcl-xL, further promoting cell survival and function [58]. For example, miR-21 targets and downregulates PTEN, a negative regulator of the PI3K/AKT pathway, leading to enhanced cell survival and proliferation [58]. All of these mechanisms may have relevant potential in mitigating gonadotoxicity and consequent premature ovarian insufficiency induced by anticancer treatments by enhancing the viability and function of granulosa cells, thus providing additional benefits in the context of aging and oxidative stress.

The results of in vivo studies in animal models also have provided significant insights into this topic. The study performed (Song et al.) in rat models of POI induced by 4-vinylcyclohexene diepoxide reveals that MenSC-derived EV exert their protective role through the modulation of the SMAD3/AKT/MDM2/P53 signaling pathways [54,55]. Specifically, these EV deliver a key multifunctional glycoprotein, thrombospondin-1 (THBS1), to target cells. THBS1, upon internalization, initiates a cascade of events leading to the inhibition of SMAD3 phosphorylation and subsequent activation of the AKT/MDM2/P53 axis. This orchestrated signaling cascade culminates in the inhibition of granulosa cell apoptosis and restoration of ovarian function [53,54]. Their results highlight the multifaceted mechanisms by which MenSC-derived EV may confer cytoprotective effects on granulosa cells, ultimately improving premature ovarian insufficiency.

In rat models of POI induced by cyclophosphamide and busulfan, Zhang et al. observed that the group of treated rats with concentrated MenSC-derived EV after induced POIS presented a significant increase in ovarian weight, follicle counts at various developmental stages (primordial, primary, secondary, and antral follicles), increased serum levels of estrogens and AMH, lower caspase-3 activity in granulosa cells, histologically reduced ovarian atrophy and fibrosis, upregulation of anti-apoptotic Bcl-2, downregulation of pro-apoptotic Bax, activation of PI3K/AKT pathway, decreased levels of p53, and increased ovarian expression of VEGF, compared to untreated POI rats [8,53]. These results highlight the multiple mechanisms by which MenSC-derived EV positively influence ovarian regeneration.

(a)Isolation and characterization of EV derived from MenSC

The following detailed comprehensive approach to isolating MenSC-derived EV is based on the most recent literature on this topic [44,60,61]. The isolation of EV from MenSC encompasses a meticulous protocol, beginning with the sterile collection of menstrual blood using a menstrual cup or similar device, followed by transfer to tubes containing an anticoagulant. The menstrual blood is diluted with phosphate-buffered saline (PBS) and subjected to Ficoll-Paque gradient centrifugation to isolate mononuclear cells. These cells are then cultured in DMEM/F12 medium supplemented with 10% fetal bovine serum (FBS) and antibiotics until they reach 70–80% confluence, with media changes every 2–3 days.

For EV collection, the cultured MenSC are incubated in EV-depleted media (prepared by ultracentrifuging FBS at 100,000 g for 16 h) when they reach the desired confluence. After 48–72 h of incubation, the conditioned media is harvested and subjected to a series of centrifugation steps to eliminate cells and debris, followed by filtration through a 0.22 µm filter. The filtered supernatant is then ultracentrifuged at 100,000 g for 70 min at 4 °C to pellet the EV, which are subsequently washed and resuspended in PBS [44,60,61].

(b)Characterization of the isolated EV involves several advanced techniques.

Nanoparticle tracking analysis (NTA) is employed to determine the size distribution and concentration of the EV, while transmission electron microscopy (TEM) is used to visualize their morphology and confirm their size. Flow cytometry can also be utilized to analyze specific surface markers of EV, such as CD63, CD81, and CD9. Maintaining stringent sterile conditions throughout the procedure is crucial to avoid contamination, and the use of EV-depleted FBS is essential to prevent contamination from bovine EV [44,60,61]. There are safety concerns regarding EV derived from MenSC. 

Exosomes derived from MSC, including those from MenSC, address the safety concerns of stem cell therapy because of their low immunogenicity in comparison to the MSC themselves, besides lacking the potential to differentiate into adult cell lineages or tumor cells. Moreover, most preparations of MSC-derived EV are characterized according to the Minimal Information for Studies of EV (MISEV2014), which was updated and expanded to MISEV2018. These guidelines, published by the International Society for Extracellular Vesicles, recommend specific criteria for the definition and classification of EV [4,6]. In Table 2, we summarized and compared the main aspects regarding the safety profiles of the biotherapies for POI mentioned in this review in the context of cancer survivors. However, the long-term impact of MenSC-derived EV is a crucial consideration in regenerative medicine that requires further investigation. Research on the sustained therapeutic effects of these EV is limited. Presently, the methods for purifying and enriching MSC-derived EV, such as ultracentrifugation, tandem filtration, and polyethylene glycol precipitation, are based on techniques initially developed considering the production of viruses or virus-like particles [4,36]. Consequently, if any viral-related products including lentiviral and adenoviral vectors used in gene editing are present in the conditioned medium or recipient cells, they could be co-enriched in the final exosome preparation, creating potential safety risks [4,6,36]. Additionally, EV are rich in miRNA, which may contribute to the instability of nucleic acid chains or induce structural changes in tissues, leading to complications. Therefore, ensuring the stability of the purity and physiological function of MenSC-derived EV remains challenging.

## 5. Conclusions

The therapeutic potential of MenSC-derived EV in premature ovarian insufficiency is underpinned by their ability to maintain genomic stability, exhibit immunomodulatory effects, and facilitate tissue regeneration. These vesicles, primarily exosomes, might mitigate ovarian dysfunction through mechanisms including the modulation of the SMAD3/AKT/MDM2/P53 pathway, enhancement of ovarian angiogenesis, and reduction in granulosa cell apoptosis. However, the current research is limited, and the methods for EV purification and enrichment generate potential risks due to co-enrichment of viral products and miRNA.

Therefore, while MenSC-derived EV hold promise in ovarian function regeneration, extensive basic and clinical research is imperative to fully understand their therapeutic potential and to ensure their safe and effective implementation in clinical practice. Future studies should focus on optimizing purification techniques, elucidating long-term effects, and validating clinical efficacy to pave the way for the therapeutic use of MenSC-derived EV in regenerative medicine, particularly in ovarian regeneration.

## Figures and Tables

**Figure 1 ijms-25-08468-f001:**
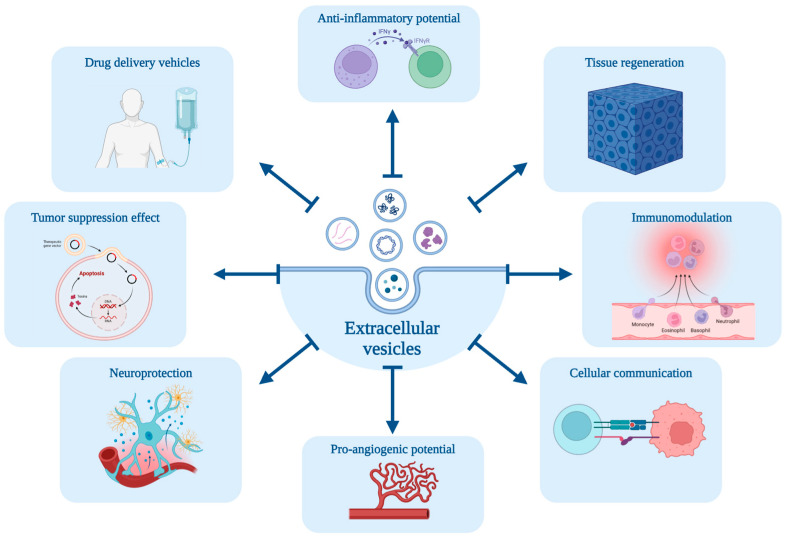
Schematic diagram of the potential properties mediated by the extracellular vesicles’ cargo of biomolecules, including anti-inflammatory potential, tissue regeneration, immunomodulation, cellular communication, pro-angiogenic potential, neuroprotection, tumor suppression effect and drug delivery vehicles. Created with BioRender.com.

**Figure 2 ijms-25-08468-f002:**
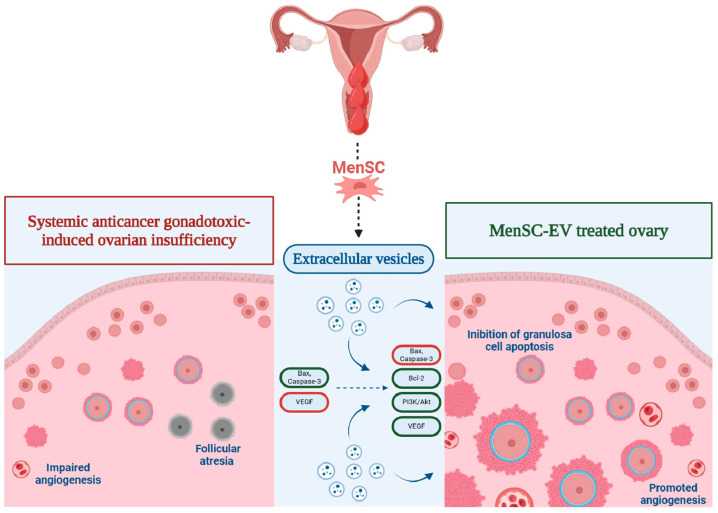
Schematic representation of the different mechanisms of menstrual blood-derived stem cell (MenSC)-derived extracellular vesicles (EV) involved in the treatment of premature ovarian insufficiency (POI). Created with BioRender.com.

**Table 1 ijms-25-08468-t001:** The potential protectants against the damaging effects of systemic anticancer treatment drugs on the ovary. PMF, primordial follicle.

Damaging Effects of Chemotherapy	Potential Protectants
Accelerated PMF activation	AMH Crocetin Immunomodulator AS101 Melatonin, Ghrelin mTOR inhibitors
Follicular atresia	Antioxidants Bortezomib Dexrazoxane G-CSF LH Imatinib Melatonin, Ghrelin Sphingolipids
Damage to ovarian vasculature	G-CSF Sphingolipids
Direct loss of PMF	ATM, ATR inhibitors CK1, CHK2 inhibitors LH Sphingolipids Tamoxifen Tyrosine kinase inhibitors
Ovarian inflammation	Tamoxifen

**Table 2 ijms-25-08468-t002:** Comparison of the safety profiles of platelet-rich plasma (PRP), mesenchymal stem cells (MSC), and MSC-derived exosomes in the context of cancer survivors.

Aspect	Platelet-Rich Plasma	Mesenchymal Stem Cells	MSC-Derived Exosomes
**Source**	Autologous (patient’s own blood)	Bone marrow Adipose tissue Umbilical cord blood	Harvested from culture medium of MSC
**Tumorigenic** **Potential**	Low risk of tumor recurrence	Potential to influence tumor growth; can promote or inhibit cancer depending on the microenvironment and source	Lower risk of tumorigenesis compared to whole MSC; potential to influence tumor biology
**Immunogenicity**	Minimal risk of immunogenic reactions or disease transmission	Risk of immune rejection or graft-versus-host disease, particularly with allogeneic MSC	Reduced risk of immune rejection or adverse immune responses compared to whole MSC
**Common Side Effects**	Pain at injection site Infection Local inflammation	Infection Thrombosis Ectopic tissue formation	Inflammatory responses (less severe than those associated with cellular therapies)
**Safety Summary**	Considered safe with minimal immunogenic risks; favorable for cancer survivors	Higher risk due to potential tumorigenicity and immunomodulatory effects	Safer than whole MSC with lower immunogenic and tumorigenic risks; long-term effects and optimal dosing require further study

## Data Availability

The original contributions presented in the study are included in the article. Further inquiries can be directed to the corresponding author/s.

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
