# Peer review of "Menstrual Blood Stem Cells-Derived Exosomes as Promising Therapeutic Tools in Premature Ovarian Insufficiency Induced by Gonadotoxic Systemic Anticancer Treatment"

_ijms, 2024, doi:10.3390/ijms25158468_

Round 1

Reviewer 1 Report

Comments and Suggestions for Authors

The problem of therapy of OI is very interested. I recommend to made additional comparison of different types of biotherapy, including safety  and efficacy of implementation of PRP, MSCs, EV from MSCs.

The results of experiment in vitro is reasonable to be added regarding these questions.

The problem of ovarian insufficiency (OI) is well known in patients treated due to cancer. The authors of review made an attempt to describe the possibilities to prevent OI by using several pathways. Among them usage of PRP, MSCs, EV of MSCs looks as perspective and effective. These ideas are based predominantly on results of experimental research in animals. Authors do not present the confirmations of direct or indirect effects of PRP, MSCs, EV of MSCs on estrogen secretion in vitro. Other questions are safety of these types of therapy in cancer-bearing animals and possibilities for patients. It is also of interest to compare these types of biotherapy regarding efficiency and safety. Of cause uploaded problem is very interested. So it is reasonable to discuss it in International Journal of Molecular Sciences.

Reviewer 2 Report

Comments and Suggestions for Authors

This study presents an intriguing manuscript demonstrating the efficacy of MenSC-derived EVs in the pathophysiology of "premature ovarian insufficiency induced by gonadotoxic systemic anticancer treatment".

However, I have the following concerns, and addressing them would make the manuscript more interesting to a broader audience.

#1 Please include and discuss more studies focusing on blood cytokine levels and the immunomodulatory capacity of exosomes derived from human menstrual blood stem cells (MenSC).

The manuscript describes the effects of exosome administration on miRNA, signaling pathways, and surface antigens, but lacks detailed information on changes in blood cytokine levels, immunomodulatory capacity, and inflammation control. While these aspects are mentioned in Figure 1, please discuss relevant studies in the text.

As indicated by the authors in reference 7, there are clinical reports on the efficacy of MenSC-derived exosomes. This study discusses the immunomodulatory properties of MenSC-derived exosomes, but unfortunately does not mention their immunomodulatory capacity.

Ref 7: https://stemcellres.biomedcentral.com/articles/10.1186/s13287-023-03413-5

#2 Please describe the method for isolation of MenSC-derived EVs.

Comments on the Quality of English Language

N/A

Round 2

Reviewer 2 Report

Comments and Suggestions for Authors

I am pleased to have had the opportunity to participate in the review of this manuscript. The authors have appropriately addressed the revisions.

I have no further comments, and I believe these revisions have made the manuscript more interesting to a broader audience.

Comments on the Quality of English Language

N/A

Author Response

Comments 1: "I am pleased to have had the opportunity to participate in the review of this manuscript. The authors have appropriately addressed the revisions. I have no further comments, and I believe these revisions have made the manuscript more interesting to a broader audience."

Response 1: We would like to thank you very much for your thoughtful and positive review of our manuscript. We greatly appreciate your feedback and are pleased to hear that you believe the revisions have enhanced the manuscript's appeal to a broader audience. Your constructive comments were invaluable in guiding our revisions, and we are grateful for your time and effort in reviewing our work.